# Volatile Composition and Sensory Profile of Lactose-Free Kefir, and Its Acceptability by Elderly Consumers

**DOI:** 10.3390/molecules27175386

**Published:** 2022-08-24

**Authors:** Jaroslawa Rutkowska, Agata Antoniewska-Krzeska, Anna Żbikowska, Patricia Cazón, Manuel Vázquez

**Affiliations:** 1Institute of Human Nutrition Sciences, Faculty of Human Nutrition, Warsaw University of Life Sciences, Nowoursynowska st.159c, 02-776 Warsaw, Poland; 2Institute of Food Sciences, Department of Food Technology and Assessment, Division of Fat and Oils and Food Concentrates Technology, Warsaw University of Life Sciences (WULS-SGGW), Nowoursynowska st.159c, 02-776 Warsaw, Poland; 3Department of Analytical Chemistry, Faculty of Veterinary, University of Santiago de Compostela, 27002 Lugo, Spain

**Keywords:** lactose-free kefir, volatile compounds, sensory attributes, just-about-right scale, consumer preferences, elderly subjects

## Abstract

Lactose-free products are crucial in the diet of lactose-intolerant elderly consumers, one of them being kefir due to its unique chemical composition and diversity of valuable microflora. The study aimed at determining the volatile compound profile and the corresponding sensory attributes of lactose-free kefir (LFK) as compared with the traditional one (TK). The perception of main sensory attributes and hedonic acceptability of LFK by elderly were also studied. The LFK contained two times more ketones, especially 3-hydroxy-2-butanone and 2,3-butanedione, that probably contributed to the high intensity of creamy aroma. A substantial share of acetic acid in LFK was not associated with high intensity of sour aroma, probably being masked by the creamy aroma, perceived as dominating. LFK was sensed as sweeter and more milky than the traditional one. The intense sweet taste of LFK was due to higher amounts of glucose and galactose than in TK, and was perceived as “just about right” by 63% of elderly subjects in the just-about-right (JAR) scale. The lower acidity of LFK than that of TK, assayed both instrumentally and by sensory assessment, was highly appreciated by 73% of elderly subjects as “just about right” in JAR scale. These two taste attributes dominated in liking the lactose-free kefir by elderly subjects.

## 1. Introduction

Kefir is a fermented, refreshing milk beverage, traditionally prepared by inoculation of raw milk with kefir grains. Kefir grains contain diverse species of lactic acid bacteria (Lactobacillus, Lactococus, Leuconostoc, and Streptococus), yeasts (Kluyveromyces, Candida, Saccharomyces and Pichia), acetic acid bacteria, and mycelial fungi in a protein/polysaccharide matrix [1,2]. Lactic acid bacteria and yeasts exist in a complex symbiotic relationship and are responsible for alcoholic and lactic fermentation, respectively [3]. Kefir contains more than 50 species of probiotic microorganisms, acting beneficially in the gastrointestinal tract via adhesion to the intestinal mucus and interference with pathogenic bacteria [2,4].

The microbial fermentation of kefir produces many valuable organic compounds, such as bioactive peptides, amino acids, exopolysaccharides, bacteriocins, antibiotics, hydrogen peroxide, vitamins (B_1_, B_12_), and calcium [2,4]. These compounds may act independently or interact, bringing about diverse health benefits attributed to kefir consumption [5]. For example, the synergistic effect of carbonyl compounds, histone, and cathelicidin with organic acids was recently reported [6]. Many authors observed antimicrobial activity of kefir against enteric bacterial pathogens, including *Escherichia coli*, *Listeria monocytogenes*, *Salmonella enterica*, *Salmonella* species (*Typhimurium* and *Arizonae*), *Shigella flexneri*, *Yersinia enterocolictica*, *Staphyloccocus aureus*, and *Enterococcus faecalis* [6,7,8,9]. Due to the potential health benefits conferred by probiotic microorganisms and/or their metabolites, the consumption of kefir is popular in Europe, Asia, and South and North America [10].

Microbial species in kefir grains also play a crucial role in generating volatile compounds that contribute to the formation of the unique flavor properties of traditional kefir [11]. The use of gas chromatography-mass spectrometry (GC-MS) enabled detecting numerous volatile compounds in the traditional kefir, mainly carboxylic acids, alcohols, aldehydes esters, and ketones [11,12,13,14,15]. Acetaldehyde, ethanol, acetoin, diacetyl, and carbon dioxide are major compounds that contribute to the creamy, refreshing, and fermented aroma. Lactic acid provides a slightly sour taste of kefir [16,17].

Olfactory and taste perceptions get reduced with aging [18]. Since odor and taste perceptions strongly depend on the volatile components of foods, diminished olfactory function decreases their perception of elderly consumers [18]. It was found that the elderly have higher optimum preferred flavor concentrations than young people [19]. Aging is also associated with digestion dysfunctions and absorption of nutrients [20]. Some conditions causing maldigestion or malabsorption seem to be more common in elderly patients [20]. These include exocrine pancreatic insufficiency, small intestinal bacterial overgrowth, enteropathies, vascular diseases, diabetes, and certain intestinal infections [20,21]. Changes of gut microflora in the elderly appear to bring about a reduction of beneficient bacteria (Lactobacilli and Bifidobacteria), and an increase of potentially pathogenic species [22]. Many researchers reported that some probiotic strains may help to keep health in old people, suggesting both health and cost-saving benefits in offering fermented dairy products. These benefits include establishment of balanced intestinal microflora; improving colonization resistance and/or prevention of diarrhea; reduction of fecal enzymes; reduction of serum cholesterol; reduction of potential mutagenes; reduction of lactose intolerance; synthesis of vitamins; and predigesting of proteins [22]. Thus, nutritive and rich in microbial species kefir seems to be appropriate beverage for the elderly. However, lactose intolerance restricts the consumption of traditional kefir by the elderly. Lactose intolerance is a gastrointestinal disorder characterized by, e.g., abdominal pain, bloating, flatulence, and diarrhea [23]. About 70% of the world’s population is β-galactosidase deficient, that enzyme being responsible for lactose hydrolysis [24]. A significant decrease in lactose intolerance is observed in people over 74 years of age, which may lead to calcium and vitamins D and B deficiency, and osteoporosis [25].

Milk fermentation into kefir reduces the lactose content in milk. Nevertheless, kefir still contains significant amounts of intact disaccharide [26]. Thus, lactose-free kefir, possessing nutritional and functional advantages typical of traditional kefir, but being free of the inadvisable compound-lactose, is an appropriate fermented milk beverage, recommendable for the elderly.

Volatile and sensory characteristics of the traditional kefir were thoroughly studied [11,13,14,27,28,29,30]. The microflora diversity of kefir grains is responsible for the formation of kefir and its aroma. Duran et al. [14] showed that *Lactobacillus kefiri* was highest active in acetic acid production and in citrate and lactose consumption, important for organic acid content and aroma formation, while *Lactobacillus parakefiri* contributed to the formation of some volatiles (acetaldehyde, ethyl hexanoate, ethyl octanoate) responsible for the desired kefir aroma profile [14]. However, to date, the information on the volatiles and on the sensory profiling of lactose-free kefir is scarce, and to the best of our knowledge, this is the first report on the volatile components and corresponding sensory attributes of the lactose-free kefir. Having in mind the needs of elderly people with lactose intolerance, their perception of main sensory attributes (sweetness, sourness, refreshing effect) and hedonic acceptability of lactose-free kefir by elderly were studied.

## 2. Results and Discussion

### 2.1. Proximate Composition

The values of pH and of acidity of traditional kefir were in accordance with those reported for kefir manufactured under similar conditions and stored for 7 days (Table 1) [12,31]. Lactose-free kefir had lower acidity than traditional kefir, probably due to differences in sugar profile [32].

Both types of kefir had similar protein content, as reported by Arslan [33], and similar fat content (1.50 g/100 g), which is common for commercial kefir in Poland (Table 1).

As expected, the two types of kefir differed significantly in the contents of major carbohydrates (Table 1). Lactose content in the traditional kefir (about 3.6 g/100 g) was in accordance with literature data of kefir stored for 7 or 8 days, while lactose level in the lactose-free kefir samples amounted to 0.1 g/100 g [31,34,35]. For a food product to be considered lactose-free, the level of lactose must not exceed 0.1 g/100 g, according to the recommendation of the European Food Safety Agency [36]. As expected, lactose-free kefir contained higher amounts of glucose and galactose than traditional kefir. Similar contents of glucose and galactose were reported by Ohlsson et al. [34] in lactose-free commercial kefir in Sweden.

Conventional microorganism enumeration was conducted after milk fermentation (on 1st day) and on 7th day storage (Table 1 and Appendix A). As reported in papers describing microflora composition of traditional kefir [31,37,38], increases in bacteria and yeast genera were noted for both types of kefir after 7 days of cold storage. The counts of *Lactobacillus* spp. and *Leuconostoc* spp. were slightly higher in traditional kefir than in the lactose-free one, but the differences were not significant (Table 1). Lactose-free kefir differed from the traditional one by significantly higher counts of *Lactococcus* spp. (Table 1). These differences may be explained by the presence of lactose in traditional kefir, lactose being a better substrate than glucose for fermentation provided by that species [39]. The differences in counts of lactic acid bacteria between the two types of kefir were reflected in their pH and acidity values (Table 1). Stored traditional kefir contained higher yeast counts (4.54 log CFU/mL) than the lactose-free one (3.70 log CFU/mL). Results of yeast enumeration in traditional kefir are in agreement with other reports [37,40].

### 2.2. Profile of Volatile Compounds

Application of the SPME/GC/MS method enabled determining and identification of volatile compounds in two types of kefir which belonged to alcohols, acids, aldehydes, ketones, terpenes, and others groups. The lactose-free kefir contained a slightly lower number of volatile compounds than the traditional kefir (28 and 31, respectively) (Table 2). A similar number of volatile compounds in the traditional kefir was reported elsewhere [11]. Semi-quantitative determination revealed also some interesting differences in the relative abundance of volatile compounds of two types of kefir (Table 2).

Acid-based volatiles are important for fermented milk products, such as kefir and yoghurt [14]. Both types of kefir significantly (*p* < 0.05) differed in the acetic acid content (30.30% and 49.18% in the lactose-free and traditional kefir, respectively). The presence of acetic acid in fermented beverages could be attributed to heterofermentative lactic acid and acetic acid cultures present in kefir grains microflora [3]. Acetic acid may contribute to acidic, vinegar, and pungent odor notes of kefir [14]. Heterofermentative species convert lactose into lactic acid, acetic acid, ethanol, and CO_2_ [41]. Acetic acid may have resulted from the metabolism of ethanol. Three other acids, butanoic, hexanoic, and octanoic, i.e., fatty acids, were detected in kefir samples in much lower quantities than acetic acid. Their contents did not differ significantly between the two types of kefir (Table 2). These acids probably originated from milk fat lipolysis [42]. Hexanoic acid can contribute to the sweaty, cheesy, and acidic aroma of kefir [11].

Ketones constituted one of the major chemical families of volatile compounds of lactose-free kefir; among the six identified ketones, the 3-hydroxy-2-butanone (acetoin) and 2,3-butanedione (diacetyl) dominated, their content being about 6- and 3-fold higher, respectively, than in the traditional kefir (Table 2). Both compounds are derived from pyruvate, the product of sugar (glucose or lactose) and citrate metabolism [15]. Most Lactobacillus species are able to co-metabolize citrate in the presence of another energy source, such as glucose or lactose, which resulted in the production of the C4 aroma compounds [39]. We suppose that the synthesis of C4 aroma compounds was more effective in lactose-free kefir than in the traditional one, probably due to a higher level of glucose in the lactose-free kefir (Table 1). Some citrate-positive lactic acid bacteria strains, e.g., *Leuconostoc* spp., are also known to produce ketone compounds from citrates, e.g., 3-hydroxy-2-butanone (acetoin) and 2,3-butanedione (diacetyl) [14]. Much higher contents of those ketones in the lactose-free kefir than in the traditional one (Table 2) may thus have been due to higher counts of *Leuconostoc* spp. in the former (Table 1). As was previously stated, 2,3-butanedione (diacetyl) may be easily converted to 3-hydroxy-2-butanone (acetion) by the diacetyl reductase, probably resulting in the higher domination of 3-hydroxy-2-butanone (acetion) compared with 2,3-butanedione (diacetyl) in the lactose-free kefir [15,43]. These two compounds are considered the main contributors to the “buttery” flavor of fermented milk [15,44].

Lactose-free kefir also contained substantial amounts of 2-butanone (7.45%), the product of reduction of 2,3-butanediol, derived from acetoin [12]. 2-Butanone is a characteristic volatile active component in yoghurt and is associated with buttery and sour flavors [12]. 2-Butanone was also previously detected in substantial amounts in kefir samples and was observed from beginning and through fermentation [43].

Among the ketone compounds, only the 2-heptanone content was alike in both types of kefir (Table 2); that compound was reported as one of the key aroma compounds that contribute to “creamy, fresh, fruity” flavor of fermented milk [14,15]. The lactose-free kefir contained significantly more of all other ketones; 2-propanone was rarely detected in fermented milk beverages [14]. The abundance of the abovementioned ketones, plus 2-nonanone, resulted from lipid metabolism by microorganisms [11,14].

The two types of kefir greatly differed in the number and contents of volatile alcohols (Table 2). Ethanol in kefir is the product of yeasts, e.g., *Saccharomyces cerevisiae*, *Candida kefir*, *Kluyueromyces lactis* [29]. Higher relative abundance of ethanol in traditional kefir than in the lactose-free one was associated with higher yeast counts in traditional kefir (Table 1).

Some Lactobacillus and Lactococcus species are known to produce ethanol (and CO_2_) by the alcohol dehydrogenase-mediated acetaldehyde reduction [13,15,29]. This may explain a higher relative abundance of ethanol and lower relative abundance of acetaldehyde in the traditional kefir compared with the lactose-free one (Table 2) [15]. High ethanol share may be associated with yeast flavor present, to some extent, in the traditional kefir [29]. However, the contribution of ethanol to overall aroma and flavor is not clear in the literature. Probably, ethanol provides a complementary flavor and its higher amounts are not desirable [14].

Traditional kefir contains a substantial relative abundance of 2,3-butanediol, regarded as a reduction product of 2,3-butanedione through acetoin [13]. Some other alcohols were also detected in the traditional kefir only, resulting probably from amino acid catabolism [14,45].

The levels of aldehydes were low compared with other compounds in both types of kefir, ranging from 2.02% to 2.97%. Aldehydes in fermented dairy products derive from transamination and decarboxylation of amino acids, by Strecker degradation, or by lipid metabolism by microorganisms [15]. Similarly to diacetyl and acetoin, acetaldehyde is an important flavor compound in fermented milk beverages, resulting from pyruvate decarboxylation or via formation of the intermediate acetyl co-enzyme A by pyruvate formate lyase, or pyruvate dehydrogenase [12,16,29,43]. Acetaldehyde may bring unique flavor of fermented beverages, when present in sufficient quantity [15]. Like in other reports, acetaldehyde content was scarce in both types of kefir, maybe due to its conversion to ethanol by alcohol dehydrogenase [12,13,29]. Among the four identified aldehydes, the relative abundance of 3-methylbutanal in the traditional kefir proved two times higher than in lactose-free kefir (Table 2). High content of 3-methylbutanal in kefir samples was also reported in another study [12]. This branched aldehyde is probably the product of the Strecker degradation of leucine [12]. 3-Methylbutanal may be related to malty and cheesy flavor of kefir [14] being, however, irrelevant because of its low relative abundance in both types of kefir (Table 2).

Terpenes were represented by six compounds, the contents of some of them being significantly (*p* < 0.05) higher in the traditional vs. lactose-free kefir; however, this was irrelevant regarding aroma profile. D-limonene, a monoterpene with citrus flavor, quantitatively dominated in both types of kefir, like in other studies on traditional kefir [14,40]. The presence of α-pinene and of β-pinene should be mentioned as these were also detected in regional cheeses [46,47]. The terpenes in kefir may originate from degradation and biosynthesis of monoterpenes and sesquiterpenes by lactic acid bacteria [14]. Terpenes are detected in milk and milk products because of their wide occurrence in plants [40]. According to Kondyli et al. [47], the abundance of terpenes in cheese may indicate that milk was derived from pasture-fed animals.

Among the three unclassified volatile compounds, the presence of toluene may be mentioned, as its low content in kefir was reported by Dan et al. [15]. Substantial amounts of toluene were detected in cheese samples, especially those derived from milk of summer season, as a product of carotene degradation in fresh grass [45,46,47]. Hydrocarbons may serve as precursors of other aromatic compounds, but their contribution to the aroma is negligible due to high threshold values [47].

Although obtained results are interesting and indicated significant differences in volatile profile of two types of kefir, it should be noted that this is a preliminary experiment and further work is needed. In particular, there is the need to replicate the trial on a larger scale, with different batches of milk, and to explore the multiple effects of the technological variables and the microbial strains.

### 2.3. Sensory Assessment of Kefir

As anticipated, traditional kefir distinctly differed from lactose-free kefir in the intensity of aroma and taste attributes (Table 3), all of them differing significantly (*p* < 0.05) between both types of kefir. The intensity of sour aroma was perceived as dominating in traditional kefir, probably resulting from higher abundance of carboxylic acids, mainly acetic, (Table 2). The substantial impact of sourness in aroma profile of traditional kefir was also reported by others [48]. The sensory scores of sour aroma may be also associated with the low pH values of kefir [17]. In traditional kefir, the fermented aroma was perceived as more intense than in the lactose-free one, probably due to about 3-fold higher amounts of alcohol compounds in traditional kefir (14.23%) than in the lactose-free one (5.49%). Some authors indicated that alcohols, especially ethanol and carbon dioxide, may contribute to the formation of fermented and alcoholic aroma of traditional kefir [12,49]. Furthermore, a substantial content of acetic acid, perceived as vinegar aroma compound, may have contributed to higher scoring of fermented aroma in traditional kefir [11].

Lactose-free kefir had a much higher intensity of creamy aroma than the traditional one (scores 7.3 and 2.4, respectively). The differences in sensory scoring of milky/creamy aroma intensity of the lactose-free kefir were supposedly due to about 3–6-fold higher contents of 3-hydroxy-2-butanone and of 2,3-butanedione than in the traditional one (Table 2); these volatile compounds were reported responsible for buttery aroma of fermented milk products [15,44]. It should be noted that 3-hydroxy-2-butanone dominated in the volatile profile of lactose-free kefir. We suppose that this compound has a crucial impact on scoring aroma of the lactose-free kefir. The results of rating creamy aroma of traditional kefir are consistent with those published elsewhere; in our study, this aroma attribute was found less perceptible in the traditional kefir, as compared with sour aroma [30,48].

The fruity aroma of lactose-free kefir had a much higher intensity than the traditional one (scores 4.47 and 1.36, respectively). According to literature data, the abundance of acetaldehyde, nonanal, and terpenes may alter the fruity aroma of kefir [11]. The sourness of traditional kefir, which scored 2.5-fold higher than the lactose-free one, was due to a high content of carboxylic acids, especially acetic acid, and that may have masked the fruity aroma of traditional kefir. A similar intensity of sour taste was also reported for sheep milk kefir stored for 7 days [40].

Traditional kefir was rated as more refreshing and bitter than the lactose-free one, the bitter taste differing remarkably (scores 2.14 and 0.42, respectively). As reported by Irigoyen et al. [27], sharp acid and yeasty flavor, together with prickly sensation due to carbon dioxide produced by yeast, can be considered as the typical kefir flavor.

Contrary to traditional kefir, the taste of lactose-free kefir was rated as sweeter and milkier (Table 3). Higher sweetness of lactose-free kefir was undoubtedly due to higher contents of glucose and galactose, compared with traditional kefir (Table 1). Considering sucrose sweetness as a reference (100%), the sweetness of glucose amounts to 70–80%, that of galactose—35%, and of lactose—20% [50]. Increased intensity of sweet taste was scored in kefir manufactured from goat milk, in which lactose was enzymatically bioconverted into galactooligosaccharides (kefir contained 0.1–0.2 g/kg lactose) [51]. Enzymatic bioconversion of lactose into galactooligosaccharides (GOS) makes the product sweeter, as the sweetness of GOS is 35% [52]. We found that the more sour and bitter was kefir, the less sweet was perceived.

The intensity of milky taste of lactose-free kefir was much higher than that of traditional kefir (scores 6.19 and 3.61, respectively), maybe due to a high content of diacetyl and 2,3-butanedione (Table 2). In fermented dairy products, various compounds with four carbon atoms: diacetyl, acetoin, and 2,3-butanediol are responsible for the typical butter-like taste [53]. These compounds can be generated by glycolysis or by citrate metabolism of several lactic acid bacteria, e.g., Lactococcus and Leuconostoc species [53]. Among these C4 compounds, diacetyl is the most important flavor compound due to its low threshold. The important effect of diacetyl on the milk aroma has been known since 1929, when it was shown that the distinctive aroma of fermented milk could be sensed when the concentration of diacetyl reached 1 mg/kg [53]. Both *Streptococcus thermophilus* and *Lactobacillus bulgaricus* are able to produce diacetyl, and strains of *Lactococus lactis* subsp. *lactis* biovar. *diacetylactis* may accumulate significant amounts of diacetyl due to their high capacity to metabolize citrate. Acetoin is the reduced form of diacetyl, and its flavor is considerably weaker than that of diacetyl. However, acetoin reduces the harshness of diacetyl and contributes to the mild creamy flavor. 2,3-Butanediol is the reduced form of acetoin, contributing somewhat to the creamy or buttery attribute [53]. Unlike lactose-free kefir, traditional kefir was distinguished by highly scored sour attribute (Table 3). High intensity of sourness of traditional kefir resulted from higher counts of lactic acid bacteria than in the lactose-free kefir, which convert lactose to lactic acid [53].

Regarding mouthfeel attributes, the mouthcoat of the lactose-free kefir was rated as more intense and smoother than of the traditional one (scores 6.95 and 4.73, respectively; Table 3). This is of high importance for dairy products as it may reflect product ability to form a film coating on the tongue and palate during consumption [54]. This could also be affected by much higher ratings of milky/creamy attributes of lactose-free kefir. Interestingly, airy attribute was scored significantly higher in traditional (scores) than in the lactose-free kefir (scores 5.6 and 3.22, respectively), probably due to the refreshing sensations and slightly higher level of carbonation (data not shown). The astringent mouthfeel was found to be about two times higher in the traditional kefir samples as compared with lactose-free kefir (scores 4.38 and 2.16, respectively). This may have resulted from the sourness and fermented attributes perceived as more intense in traditional kefir (Table 3). Mouthfeel is regarded as of high importance for elderly subjects, who commonly experience taste loss and olfactory dysfunction [55,56].

### 2.4. Preferences of Lactose-Free Kefir by Elderly Consumers

The results of consumer assessment of the appropriateness of intensity of selected attributes (sweetness, acidity, and refreshing effect) by JAR scale are summarized in Table 4. The results clearly deviate from normality, as mentioned by Bayarri et al. [57].

Consumer ratings of sweetness conducted by JAR scale revealed that 63% of elderly consumers perceived intensity of sweetness as “just about right”. Only about 8% of them perceived sweetness as “far too little” or “far too much”. The majority of elderly subjects also perceived acidity as “just about right” (73%), and nobody scored acidity as “far too much”. In contrast, the intensity of refreshing effect was scored differently. About 42% of consumers perceived refreshing effect of the lactose-free kefir as “just about right” and, similarly (43%) as “too little” and “far too little” (Table 4).

On the average, female subjects rated sweet taste using JAR scale significantly (*p* < 0.01) higher than the male ones, mean values (±SD) amounting to 2.94 ± 0.78 and 2.60 ± 0.67, respectively. Thus, women perceived the intensity of sweet taste as more appropriate than men (Figure 1). The Chi-square test revealed significant (*p* < 0.05) differences in two rating categories: “Too little” (2 points) and “Just about right” (3 points).

Tourila et al.’s [58] study on fermented milk beverages revealed that male consumers preferred high sweetness of strawberry yogurt than female ones. However, Chollet et al. [59] did not show a sex effect in evaluating sweetness adequacy of flavored yogurts. These divergences in perceiving the intensity of sweetness of fermented beverages may have derived from sugar origin in various types of beverages (yoghurt and kefir). The sweetness of yoghurt is created by added sugars, while the sweetness of lactose-free kefir is due to the presence of free glucose- a product of enzymatic decomposition of lactose. Thus, lactose-free kefir was perceived as natural regarding sweetness.

No significant gender-related differences in mean values were found for either acid taste or refreshing effect, except a slight but significant (*p* < 0.05) difference in the “Too much” category for acid taste (Figure 1).

The frequencies of hedonic acceptability scores are presented in Figure 2. About 50% of elderly consumers rated lactose-free kefir as “like extremely” or “like very much”. Only 12% of them rated the lactose-free kefir more or less negatively, but nobody rated it as “dislike extremely”. The high preference of lactose-free kefir was associated with high intensity of sweetness, as indicated also by trained sensory panel (Table 3). Sweetness was reported to substantially increase the liking of fermented milk drinks [54,60].

It is worth noting that the consumers rated highest the lactose-free kefir as “just right” in all three attributes—sweet taste, acid taste, and refreshing effect, the percentages amounting to 60.9, 70.7, and 43.7%, respectively (Table 4). Similar results were reported earlier [18].

The trained sensory panel assayed intensity of sour taste lactose-free kefir as low which was regarded as appropriate on JAR scores by elderly consumers. Our study confirmed that high intensity of sour taste decreased liking of fermented beverages [54].

## 3. Materials and Methods

### 3.1. Manufacturing Kefir

Both types of kefir were manufactured in a small dairy plant, using cow milk from Polish Holstein Friesian and Siementhal breeds of high microbiological and cytological quality derived from local farms located in the Mazovian region in Poland. Two types of kefir were manufactured from the same three independent parts of milk, standardized to 1.50% fat with a cream separator, and homogenized, one part of milk being subjected to the enzymatic process to decompose lactose by β-galactosidase activity from *Kluyveromyces lactis* of the commercial enzymatic preparation GODO-YNL2 Lactase (DuPont^TM^ Danisco A/S, Brabrand, Denmark); the reaction lasted 24 h at 36 °C. Next, milk was pasteurized at 92 °C for 7 min, cooled rapidly down to 23 °C, and a lyophilized starter culture containing *Lactobacillus* spp., *Lactococcus* spp., *Leuconostoc* spp., *Streptococus thermophilus*, *Lactobacillus rhamnosus*, and kefir yeast (DuPont^TM^ Danisco A/S, Brabrand, Denmark). Kefir starter culture was added to milk at the level 0.025 g/L. The inoculated milk was distributed into polyethylene bottles and incubated in a thermostatically controlled chamber at 22–23 °C, the fermentation lasting 18 h. Samples were stored at 4 °C. From the second part of pasteurized milk, traditional kefir was manufactured using the same technological parameters and the same starter culture as in the case of lactose-free kefir. The kefirs were manufactured in three independent trials, 80 L each batch. Samples of both types of kefir were stored for 7 days and taken for analysis.

All chemical and microbial analyses were performed in triplicate for each batch. Sensory assessment by an experienced panel was conducted in three sessions, while consumer evaluations by elderly subjects were in two independent ones.

### 3.2. Proximate Composition and Physicochemical Measurements

The content of ash in kefir samples was determined according to the AOAC method [61] as described in a previous study [62]. Total fat content was measured using the Röse–Gottlieb method as reported earlier [63]. The protein content was assayed using the Kjeldahl method and multiplying by 6.38 according to AOAC procedure No: 991.20 [64]. Titratable acidity was measured using 0.1 mol/L NaOH and 10 g/L phenolphthalein (Sigma Aldrich, Poznan, Poland) solution in 950 mL/L ethanol as an endpoint indicator [54]. The pH was determined by using CP-411 pH-meter equipped with a combined electrode (Elmetron Company, Zabrze, Poland).

### 3.3. Enumeration of Microorganisms

To evaluate the microbial content of both kefir samples, 10 mL of kefir was aseptically taken, dispersed with 90 mL of sterile Ringer solution (1:9, w/v; Merck, Darmstadt, Germany), and homogenized for 1 min in a stomacher (Lab-Blender 400; London, UK). Decimal dilutions were prepared and plated in triplicates for bacterial and yeast counts.

*Lactobacilli* spp. counts were determined on de Man Rogosa and Sharpe agar (MRS; Merck 1.10660) at 30 °C under anaerobic conditions for 3 days. *Lactococcus* spp. were counted on M17 agar (Merck) at 37 °C under anaerobic conditions for 2 days. *Leuconostoc* spp. counts were determined on de Man Rogosa and Sharpe agar (MRS; Merck) incorporated with vancomycin hydrochloride (Sigma Aldrich, Poznań, Poland) at 30 °C for 3 days. For enumerating the lactic acid bacteria, samples were plated on de Man Rogosa and Sharpe agar (MRS; Merck) and anaerobically incubated at 37 °C for 3 days. Yeasts were cultured on potato dextrose agar (PDA; Merck, 1.10130; pH 3.5) with 10% added tartaric acid at 25 °C for 3 days [17,37]. The results were expressed as the logarithm colony forming units per milliliter of kefir (log CFU/mL).

### 3.4. Determination of Carbohydrates

For sample preparation and chromatographic analysis, the previously developed procedures were applied [34,65,66]. First, milk protein was precipitated by using Carrez solutions I (2.7 g K_4_Fe (CN)_6_ in 100 mL) and II (5.5 g Zn(OAc)_2_ in 100 mL), then the samples were centrifuged at 4000 rpm for 30 min at 4 °C in order to remove fat. The mixture was filtrated through a cellulose syringe filter (Agilent Captiva Premium Syringe Filter, Regenerated Cellulose, 0.45 µm, 25 mm).

The analysis was accomplished by high-performance anion-exchange chromatography with pulsed amperometric detection (HPAEC-PAD) using Thermo Scientific Dionex ICS-3000 system (Thermo Fisher, Sunnyvale, CA, USA). The PAD detector included an electrochemical cell equipped with a disposable gold electrode and Ag/AgCl reference electrode. For chromatographic separation, the following columns were used: Dionex CarboPac PA100 (250 × 4 mm) equipped with a guard column of the same stationary phase and a borate trap pre-column (50 × 4 mm).

Separation was performed following the procedure developed by Van Scheppingen et al. [66]. Gradient elution of following mobiles phases was used: A-MilliQ purified water, B-20 mM NaOH, C-500 nM NaOH, and D-100 mM NaOH + 1 M sodium acetate. All details of gradient elution and separation are presented in another study [66]. Quantification of lactose, glucose, and galactose was performed using the external standard method. Sigma-Aldrich reference sugars were used (Sigma-Aldrich, Poznań, Poland).

### 3.5. Analysis of Volatile Compound Profile

Volatile compounds in kefir samples were determined by headspace solid phase micro-extraction (SPME) coupled with gas chromatography/mass spectrometry (GC/MS) (6890N GC, 5975 MS Agilent, Santa Clara, CA, USA).

The SPME extraction of volatile compounds was performed using a manual SPME holder with fiber coated with divinylbenzene/carboxen/polydimethylsiloxane (DVB/CAR/PDMS; Supelco, Bellefonte, PA, USA), conditioned prior to use. The 5 mL kefir samples were transferred into headspace glass vials (20 mL) and hermetically sealed with septa. The vials were kept in water bath (35 °C) for equilibration (30 min). SPME extraction (40 min) was provided under constant stirring (1000 rpm) with a magnetic stirrer. After sampling, the SPME fiber was withdrawn into the needle, removed from the vial and inserted into the injector (270 °C) of the GC/MS instrument for 5 min, where the extracted volatiles were thermally desorbed directly to the column.

The volatile compounds were separated on an HP-5MS column: 30 m × 0.25 mm × 0.25 μm film thickness, 5%-diphenyl-95%-polydimethylsiloxane (Agilent, Santa Clara, USA). Chromatographic separation was conducted under the following conditions: oven temperature was held for 10 min at 38 °C, then increased up to 200 °C (4 °C/min gradient) and held for 2 min, then raised to 250 °C at 20 °C/min, and that final temperature was held for 7 min. The mass-selective detector was operated at 70 eV, and the mass range was 30–350 m/z. The data obtained from GC/MS were processed using MSD ChemStation software (Agilent, Santa Clara, USA). Volatile compounds were identified by comparing their mass spectra with those of the NIST.08 and Wiley 7th Ed (National Institute of Standards and Technology, USA) libraries and by computing retention indices relative to a series of standard alkanes (C6-C20, Kovats indexes; Sigma-Aldrich, Poznań, Poland). The quantities of volatile compounds were expressed as relative peak areas (peak area of each compound in Appendix A/total area) × l00.

### 3.6. Sensory Estimation

The sensory assessment of kefir samples was conducted by ten panelists (6 females and 4 males aged 32–45 years), experienced in the evaluation of dairy products. Prior to the assessment, the panelists were trained on sensory descriptors and participated in the selection of sensory attributes from literature data [16,27,54,67].

The panelists evaluated the intensity perceived for each attribute using an unstructured 10 cm linear scale ranging from “no intensity” to “very high intensity” according to Baryłko-Pikielna [68]. The results from analogous scale were converted to numerical values (0 to 10 units). The panelists evaluated intensity of fourteen attributes in the following order: aroma (fermented, creamy, sour, yeasty, sweet, fruity), taste (sour, milky, sweet, bitter, refreshing), and mouthfeel (mouthcoat, airy, astringent). Definitions of attributes considered in sensory analyses are presented in Appendix A. The rating was conducted three times. During sessions, the panelists were provided with water and unsalted crackers for palate cleansing. The assessments were carried out at a sensory laboratory room fulfilling the requirements of the ISO standard.

### 3.7. Consumer Acceptance of Lactose-Free Kefir

The consumer test of lactose-free kefir was done by 256 consumers (158 females and 98 males) aged 65–76 years, recruited from three Day Residences for Seniors located in Warsaw.

Consumer acceptance of lactose-free kefir was evaluated using just-about-right (JAR) scale and hedonic scale. The participants were asked to assess the sweet taste, acid taste, and refreshing effect using the JAR scale from 1 to 5 (1 = way too little, 2 = too little, 3 = just about right, 4 = too much, 5 = way too much) [69]. The overall consumer’s degree of liking was rated using a hedonic category scale ranging from 1 “dislike extremely”, to 9 “like extremely”, with the neutral point being 5 “neither like nor dislike”. The consumers tasted 100 mL samples of kefir, served at 8–10 °C in plastic cups. Background information on age, gender and use frequencies of fermented milk beverages was also collected. Before participating in the consumer session, all consumers read an information sheet about experiment and signed their informed consents. Our research protocol followed the guidelines of the Helsinki Declaration and all procedures involving human subjects were approved by the Committee on Ethics in Human Beings Research of the Institute of Human Nutrition Sciences, Faculty of Human Nutrition, Warsaw University of Life Sciences (Reference: 05/21).

### 3.8. Data Analysis

One-way ANOVA was used to identify differences between proximate composition, microbial enumerations, relative abundance of volatile compounds and sensory attributes over two types of kefir. When a significant F-value was found, additional post hoc tests with Tukey adjustment were performed. Statistical significance was set at *p* < 0.05 confidence. Numeric values of categories of the “Just about right” scale (JAR) were used to compute gender-related mean values and to compare them by Student’s *t*-test. The frequencies of those categories for male and female subjects were subjected to the Chi-square test. The level of *p* < 0.05 was considered significant. The Statistica 3.1 software (Statsoft, Inc., Tulusa, OK, USA) was used.

## 4. Conclusions

Alcohols, ketones, and acids present in the volatile profile were the main groups differentiating both types of kefir. As compared with traditional kefir, the lactose-free one contained twice more ketones, especially 3-hydroxy-2-butanone and 2,3-butanedione, which probably contributed to the high intensity of creamy aroma scored in sensory assessment. A substantial share of acetic acid in the volatile profile of lactose-free kefir was not associated with high intensity of sour aroma, probably because of masking by milky/creamy aroma, perceived as dominating.

In contrast to traditional kefir, the lactose-free one was reported as sweeter and milkier. The intense sweet taste of lactose-free kefir was due to higher amounts of glucose and galactose than in traditional kefir, and was perceived as “just about right” by 63% of elderly subjects in the just-about-right (JAR) scale. The study revealed that women perceived the intensity of sweet taste as more appropriate than men. Lower acidity of lactose-free kefir than that of traditional kefir, assayed both instrumentally and sensory, was highly appreciated by 73% of elderly subjects as “just about right” in JAR scale.

There is evidence that an appropriate intensity of sweet and sour taste influenced hedonic acceptability of lactose-free kefir by the elderly, as about half of the consumer panel scored lactose-free kefir as “like extremely” and “like very much”, so these both taste attributes revealed as crucial drivers of consumer’s liking of lactose-free kefir.

This is the first report on the volatile profile and sensory attributes of lactose-free kefir. However, further research is required to understand activity of different strains of microflora on the volatiles formation especially that which influence desired aroma profile of lactose-free kefir.

## Figures and Tables

**Figure 1 molecules-27-05386-f001:**
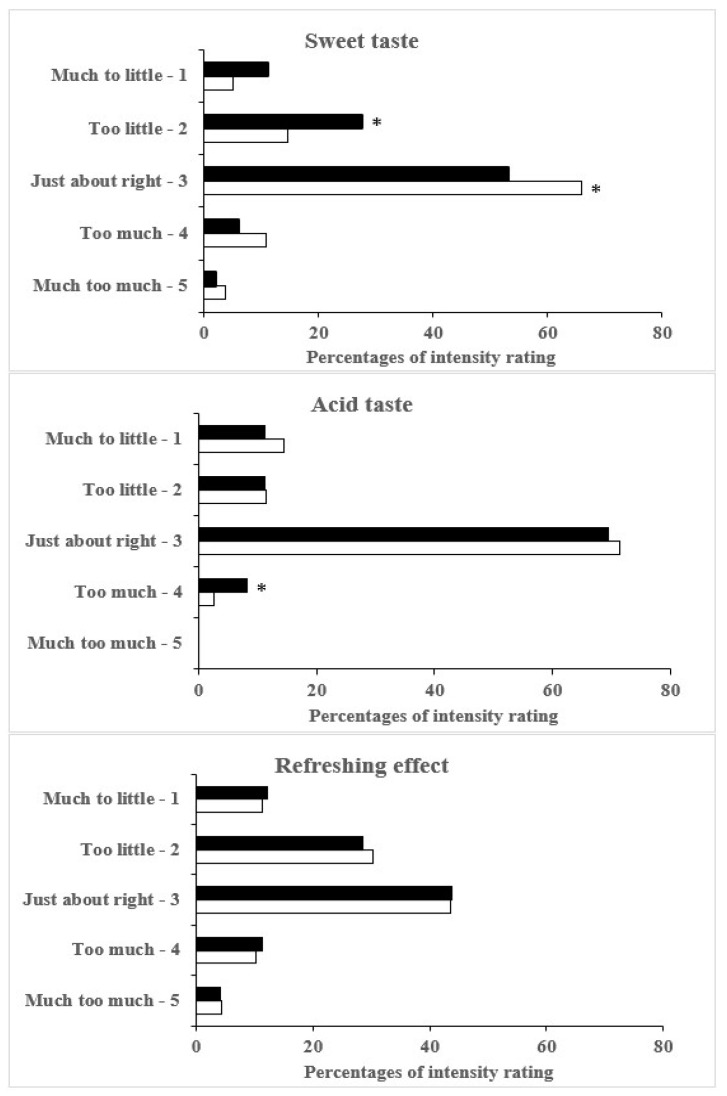
Gender effect on consumer assessment of the intensity of sweet and acid taste by using “just-about-right” scale (Much too little—1; Much too much—5). * significant (*p* < 0.05) differences between ☐ females (n = 158) and ■ males (n = 98).

**Figure 2 molecules-27-05386-f002:**
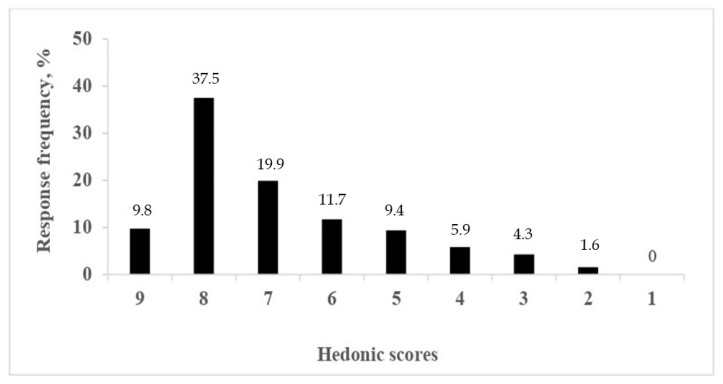
Distribution of individual preferences of lactose-free kefir (n = 256). 9—Like extremely; 8—Like very much; 7—Like moderately; 6—Like a little; 5—Neither like nor dislike; 4—Dislike a little; 3—Dislike moderately; 2—Dislike a lot; 1—Dislike extremely.

**Table 1 molecules-27-05386-t001:** Proximate composition and microbial count of kefir samples (n = 9) stored for 7 days.

Component, g/100 mL	Lactose-Free Kefir	Traditional Kefir
Proteins	3.20 ± 0.05	3.16 ± 0.03
Lipids	1.52 ± 0.02	1.55 ± 0.03
Lactose	0.01 ± 0.01	3.12 ± 0.10 *
Glucose	1.72 ± 0.10	0.35 ± 0.12 *
Galactose	2.47 ± 0.21	0.62 ± 0.09 *
Total acidity, % of LA	0.62 ± 0.08	0.86 ± 0.01 *
pH	4.82 ± 0.10	4.45 ± 0.03 *
Ash, %	0.62 ± 0.04	0.75 ± 0.02 *
Microbial enumeration, log CFU/mL		
*Lactobacillus* spp.	7.83 ± 0.09	8.06 ± 0.13
*Lactococcus* spp.	8.35 ± 0.10	7.96 ± 0.24 *
*Leuconostoc* spp.	5.64 ± 0.16	5.82 ± 0.08
LAB	8.21 ± 0.13	8.65 ± 0.20 *
Yeast	3.70 ± 0.08	4.34 ± 0.12 *

* Significantly (*p* < 0.05) different from the lactose-free kefir. LA—Lactic acid; CFU—colony-forming unit; LAB—lactic acid bacteria.

**Table 2 molecules-27-05386-t002:** Volatile compounds identified in kefir and their relative abundance (means ± SD, n = 9).

Compounds		Peak Relative Abundance (%)	
	Rt	Lactose-Free Kefir	Traditional Kefir
** *Alcohols* **			
Ethanol	1.67	1.52 ± 0.11	2.20 ± 0.16 *
Ethanethiol	1.85	1.83 ± 0.36	2.37 ± 0.24 *
1-Pentanol	4.30	0.44 ± 0.18	1.69 ± 0.05 *
3-Methyl-1-butanol	4.41	nd	1.17 ± 0.02
2,3-Butanediol	6.27	1.08 ± 0.33	5.94 ± 0.51 *
2-Pentanol	7.07	nd	0.35 ± 0.05
1-Hexanol	11.39	nd	0.25 ± 0.00
2-Heptanol	13.47	0.61 ± 0.00	0.27 ± 0.02 *
*Total*		5.49 ± 0.17	14.23 ± 0.20 *
** *Acids* **			
Acetic acid	2.58	24.99 ± 1.28	43.30 ± 1.56 *
Butanoic acid	6.78	1.27 ± 0.23	1.27 ± 0.39
Hexanoic acid	19.37	3.57 ± 0.39	4.10 ± 0.21
Octanoic acid	26.86	0.48 ± 0.05	0.52 ± 0.06
*Total*		30.30 ± 0.49	49.18 ± 0.84 *
** *Aldehydes* **			
Acetaldehyde	1.49	0.47 ± 0.05	0.22 ± 0.03 *
3-Methylbutanal	2.81	1.09 ± 0.12	2.59 ± 0.11 *
Benzaldehyde	16.80	0.08 ± 0.03	0.02 ± 0.02 *
Nonanal	23.86	0.39 ± 0.02	0.12 ± 0.02 *
*Total*		2.02 ± 0.06	2.97 ± 0.04 *
** *Ketones* **			
2-Propanone	1.77	4.35 ± 0.40	8.96 ± 0.55 *
2,3-Butanedione	2.19	7.05 ± 0.86	2.23 ± 0.23 *
2-Butanone	2.25	7.45 ± 0.05	10.93 ± 0.78 *
3-Hydroxy-2-butanone	3.73	39.26 ± 2.27	6.95 ± 0.40 *
2-Heptanone	12.74	0.69 ± 0.18	0.78 ± 0.06
2-Nonanone	23.35	0.96 ± 0.00	0.36 ± 0.02 *
*Total*		59.75 ± 0.80	30.22 ± 0.47 *
** *Terpenes* **			
α-Pinene	15.14	0.18 ± 0.01	0.35 ± 0.18 *
β-Pinene	17.54	0.29 ± 0.02	0.40 ± 0.21
3-Carene	19.37	0.22 ± 0.02	0.32 ± 0.16
*m*-Cymene	20.10	0.08 ± 0.02	0.15 ± 0.06 *
D-Limonene	20.30	0.33 ± 0.00	0.52 ± 0.37 *
Copaene	33.81	0.07 ± 0.01	0.12 ± 0.02
*Total*		1.17 ± 0.02	1.86 ± 0.15 *
** *Other compounds* **			
Toluene	5.23	0.11 ± 0.03	0.16 ± 0.01
Dimethyl disulfide	4.54	0.91 ± 0.35	0.82 ± 0.20 *
2-Methyltetrahydro-thiophen-3-one	18.21	0.24 ± 0.07	0.56 ± 0.25
*Total*		1.27 ± 0.18	1.54 ± 0.40

* Significantly (*p* < 0.05) different from the lactose-free kefir; nd—not detected; Rt—retention time.

**Table 3 molecules-27-05386-t003:** Mean values (±SD) of the intensities of aroma, taste and mouthfeel attributes of kefir assayed by trained panel (n = 10) (no intensity—0; very high intensity—10).

Attributes	Lactose-Free Kefir	Traditional Kefir
**Aroma**		
Fermented	2.62 ± 0.18	3.78 ± 0.25 *
Creamy	7.29 ± 0.32	2.41 ± 0.13 *
Sour	1.80 ± 0.16	4.55 ± 0.27 *
Yeasty	0.74 ± 0.11	1.63 ± 0.13 *
Sweet	2.88 ± 0.26	0.42 ± 0.09 *
Fruity	4.47 ± 0.35	1.36 ± 0.11 *
**Taste**		
Sour	2.81 ± 0.24	7.33 ± 0.41 *
Milky	6.19 ± 0.33	3.61 ± 0.23 *
Sweet	4.78 ± 0.36	0.67 ± 0.10 *
Bitter	0.42 ± 0.08	2.14 ± 0.12 *
Refreshing	4.67 ± 0.21	5.28 ± 0.35 *
**Mouthfeel**		
Mouthcoat	6.95 ± 0.40	4.73 ± 0.26 *
Airy	3.22 ± 0.19	5.60 ± 0.31 *
Astringent	2.16 ± 0.14	4.38 ± 0.24 *

* Significantly (*p* < 0.05) different from the lactose-free kefir.

**Table 4 molecules-27-05386-t004:** Percentages of intensity rating of the lactose-free kefir by elderly consumers (n = 256) (Much too little—1; Much too much—5).

Intensity	Attribute		
	Sweet Taste	Acid Taste	Refreshing Effect
Much too little—1	7.4	13.3	11.7
Too little—2	19.5	11.3	29.7
Just about right—3	60.9	70.7	43.8
Too much—4	9.0	4.7	10.5
Much too much—5	3.1	0	4.3

## Data Availability

Not applicable.

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
