# Peer review of "Volatile Composition and Sensory Profile of Lactose-Free Kefir, and Its Acceptability by Elderly Consumers"

_molecules, 2022, doi:10.3390/molecules27175386_

Round 1
Reviewer 1 Report
The manuscript presented how lactose-free kefir differs from traditional kefir, and examined sensory quality and volatile compounds. Some recommendations are mentioned below.
Is elderly people the scientifically and politically correct term to be used? And how old do you have to be in that group?
Isn't kefir supposed to have pH of 4.2 to 4.6? Results shown for the lactose-free kefir is beyond pH5. Is this really "kefir"? How would you define kefir? in Line 323, it says "Fermentation was ended when pH reached 4.6". why was there such a big difference from the end of fermentation to the time proximate analysis was done for pH?
line 87-88: rephrase "Some strains, eg. Lactobacillus parakefiri, were highlighted in terms of a desired aroma profile, as well as the sensory support". Hard to understand what this is supposed to mean.
line 112-113: rephrase and specify "Similar differences were reported by Ohlsson et al. [34]." what were the similar differences reported? and how were they similar to the current findings?
section 3.1: require more details of where the milk was sourced (country) and what sort of cattle breed, where the bacteria were sourced and their counts in CFU as the starter culture. Further elaboration is desired for the manufacturing step, and the volume of production per batch is required.
section 3.2: why was 6.25 used for kjeldahl calculation instead of 6.38 which is the norm for cow's milk?
Information on repeats desired for all analyses performed
line 378: "Other details are presented elsewhere [42]." what are these other details?
section 3.6: name of the ethics committee who approved the study should be mentioned
Table 2: data are presented as % volatile compounds instead of concentration. Explain how this was calculated. Is this qualitative data without standard curves and references used to calculate the exact concentration of each volatile? How can you say data are significantly different from one another when they were not properly quantified at first place?
Table 3: I thought 10 panellists were used to gather the initial data, not n=3?
Why were the other physicochemical properties not tested? Eg. colour, viscosity etc?
Statistical analysis could be improved - what was performed in the study were too basic. Could there have been gender effect?
Author Response
Dear Editor,
Replies to Reviewer 1 is attached in a separate document.

Reviewer 2 Report
This manuscript evaluated the volatile composition (SPME-GC-MS) and sensory profile (trained panel) of lactose-free kefir, compared with the traditional kefir, and its acceptability by elderly consumers (JAR). This study is novel and of importance and interest to the field.
One area that could be improved is the method of volatile quantification. The study reported the relative abundance of the volatiles while it is unknown how much of the volatiles are in there or how many of the volatiles actually had an impact on the odor (or had a concentration higher than its odor threshold). For example, the total volatile compounds from one type of kefir could be much higher than the other. In that case, the results cannot be directly compared to the percentage of relative abundance. The results discussion also needs to take this limitation into consideration.
Table 2. Indicate clearly that the volatile compounds are expressed as relative abundance.
Tables 3 and 4. Indicate the scale of the rating in the table captions.
L378: Specify how the results are expressed.
Author Response
Dear Editor,
Replies to Reviewer 2 is attached in a separate document.

Author Response
Dear Editor,
Replies to Reviewer 3 is attached in a separate document.

Round 2
Reviewer 3 Report
Quality of presentation is low. There are only two samples with only volatile components and sensory attributes ouptus without experimental plan, optimization of techological process of manufacturing kefir, and the main topic has not been addressed from a technological point of view.
The results and discussion require more scientific explanations.
Results and disccusion and materials and methods in round 2 are little bit improved for paragraph Enumeration of microorganisms.
Unfortunately, cannot be advised for publication in Molecules. The paper is well written, but the number of outputs (experiments) for journal with a high impact factor is not enough. Volatile components and sensory attributes of the only two different type of kefir aren’t enough for journal with a high impact factor. Unfortunately, cannot be advised for publication in Molecules. Maybe to try to publish on national level journal.
Author Response
Replies to Reviewer is attached in a separate document
